# Three-Dimensional Cell Culture Models to Investigate Oral Carcinogenesis: A Scoping Review

**DOI:** 10.3390/ijms21249520

**Published:** 2020-12-14

**Authors:** Ravi Teja Chitturi Suryaprakash, Omar Kujan, Kate Shearston, Camile S. Farah

**Affiliations:** 1UWA Dental School, The University of Western Australia, Nedlands, WA 6009, Australia; raviteja.chitturisuryaprakash@research.uwa.edu.au (R.T.C.S.); kate.shearston@uwa.edu.au (K.S.); 2Australian Centre for Oral Oncology Research and Education, Nedlands, WA 6009, Australia; camile@oralmedpath.com.au; 3Oral, Maxillofacial and Dental Surgery, Fiona Stanley Hospital, Murdoch, WA 6150, Australia; 4Head & Neck Pathology, Australian Clinical Labs, Subiaco, WA 6008, Australia

**Keywords:** spheroids, organotypic raft cultures, organoids, oral mucosa, oral epithelial dysplasia, oral squamous cell carcinoma, oral cancer

## Abstract

Three-dimensional (3-D) cell culture models, such as spheroids, organoids, and organotypic cultures, are more physiologically representative of the human tumor microenvironment (TME) than traditional two-dimensional (2-D) cell culture models. They have been used as in vitro models to investigate various aspects of oral cancer but, to date, have not be widely used in investigations of the process of oral carcinogenesis. The aim of this scoping review was to evaluate the use of 3-D cell cultures in oral squamous cell carcinoma (OSCC) research, with a particular emphasis on oral carcinogenesis studies. Databases (PubMed, Scopus, and Web of Science) were systematically searched to identify research applying 3-D cell culture techniques to cells from normal, dysplastic, and malignant oral mucosae. A total of 119 studies were included for qualitative analysis including 53 studies utilizing spheroids, 62 utilizing organotypic cultures, and 4 using organoids. We found that 3-D oral carcinogenesis studies had been limited to just two organotypic culture models and that to date, spheroids and organoids had not been utilized for this purpose. Spheroid culture was most frequently used as a tumorosphere forming assay and the organoids cultured from human OSCCs most often used in drug sensitivity testing. These results indicate that there are significant opportunities to utilize 3-D cell culture to explore the development of oral cancer, particularly as the physiological relevance of these models continues to improve.

## 1. Introduction

Mucosal malignancies of the epithelial lining of the lip and oral cavity are termed oral squamous cell carcinomas (OSCCs) and are a major cause of mortality and morbidity worldwide [1]. Tobacco and alcohol seem to be the major risk factors for OSCC, but recent evidence also shows an association of other factors, such as inflammation and oral microbiome, in oral cancer. There seem to be several inflammatory mediators, salivary proteins, and oral microbiota that are common between inflammatory diseases of the oral cavity, such as periodontitis and oral cancer [2,3,4]. Despite some advances in diagnosis and treatment, the 5-year survival rate for OSCC is less than 50% across all stages, and has not greatly improved for the past few decades [5,6,7]. Survival rates are significantly higher in OSCC identified and treated at earlier stages [8]. In order to provide the best possible care for patients with OSCC, a detailed understanding of the molecular process of oral carcinogenesis and physiologically relevant models for testing potential drug treatments are required. This can facilitate the identification of potential biomarkers of malignant transformation and can accelerate the development of novel chemo-preventative agents for this disease.

Traditionally, cell culture experiments are performed with cells grown in a two-dimensional (2-D) fashion on tissue culture plastic. Culturing cells in 2-D significantly changes the gene expression profile and does not represent the complex cell–cell or cell–matrix interactions seen in normal or cancerous tissues [9]. As a result, there have been significant challenges in translating drugs tested and optimized in 2-D cultures to the clinic, where they frequently demonstrate reduced sensitivity [10]. When candidate drugs are tested in 2-D culture, the cells are exposed to a homogenous environment containing growth factors, cytokines, oxygen, and other nutrients, as well as various concentrations of the drug. In contrast, in vivo tumor cells are present in a three-dimensional (3-D) solid structure and are exposed to varying gradients of biochemical molecules and signals. 3-D cell culture models are a more physiologically relevant mimic of in vivo tumor tissues and can be used to more accurately assess the efficacy and likely dosage of anticancer drugs [11].

Oral carcinogenesis occurs as a multistep process, where mutations accumulate over a variable period of time, but despite significant developments, our understanding of the natural course of the disease remains incomplete [12]. With the global move towards personalized medicine gaining significant momentum, it is important for cancer researchers to understand the microenvironment of normal and dysplastic tissues of the oral cavity as well as the tumour microenvironment (TME) of OSCC. For these purposes, cell culture techniques based on 3-D technology are promising avenues to pursue, as they are more physiologically representative of the human TME than traditional 2-D methods or in vivo animal models.

A number of 3-D cell culture technologies have been developed, the most important of which include spheroids, organotypic raft cultures, and organoids. The basic concept of generating spheroids is to encourage cells to interact with each other rather than allowing them to attach to a plastic surface, which allows them to form 3-D cell structures. Spheroids are attractive models for cancer research as they recapitulate the various cell–cell signaling pathways and growth kinetics of cancer cells [13,14]. They are the simplest technology as far as technical aspects are concerned among various 3-D cell culture methods. The methods by which these spheroids are generated are beyond the scope of this review and have been discussed in detail by other authors [15,16]. In general, spheroid techniques either use only the culture media and a method to avoid adherence, or are grown on matrices in order to promote spheroid growth. There are a number of suspension-based techniques to generate spheroids, which have been summarized in Figure 1. The suspension-based models mimic cell–cell interactions in tumors and encourage spheroid formation, but not all cells will form spheroids using these methods. Some cells typically require matrices to form spheroids, which provide both cell–cell as well as cell–matrix interactions. There are generally two techniques (Figure 2) by which matrix-derived spheroids can be grown: 1) cells above the matrix and 2) cells embedded within the matrix. For this purpose, various materials have been used, including both natural and synthetic matrices. Naturally derived matrices are more representative of the TME as they resemble the matrix component of solid tumors; the most commonly used being Collagen or Matrigel. Matrigel is a reconstituted basement membrane matrix derived from Engelbreth–Holm–Swarm (EHS) mouse sarcoma cells, which has been known to display batch to batch variability [17]. Other examples of natural matrices used in 3-D culture are summarized in Table 1. A range of synthetic matrices have also been used to generate spheroids, including polyethylene glycol (PEG), polylactic-co-glycol (PLGA), and poly-ε-caprolactone (PCL), which are more uniform but less physiologically suitable compared to natural matrices [18]. One of the advantages of using the synthetic matrices is that the growth factors/cytokines/additives required for culture can be customized as compared to Matrigel or any of the naturally derived matrices, which may contain undefined growth factors.

Organotypic raft culture is another important 3-D model representing epithelial tissues as it recapitulates the differentiation and stratification of keratinocytes [25]. A number of culture models have been developed from various cells of the oral cavity, including normal, dysplastic, and OSCC cells/cell lines, and have been used to study viral oncogenesis of head and neck squamous cell carcinoma and other viral diseases associated with the oral mucosa [26,27]. In this model, stratification is achieved by culturing keratinocytes over a scaffold of dermal equivalents, such as de-epidermized skin, collagen, or Matrigel with/without fibroblasts in an air–liquid interface. As far as cancerous tissues are concerned, these types of 3-D cultures have been principally used as invasion models. For this purpose, human uterus leiomyoma tumor-derived discs have also been quite popular [28].

Organoids are the latest development in 3-D cell culture technology and can be best described as miniaturized versions of a patient’s organs in culture. They include multiple cell types as seen in vivo and organize and function in a fashion similar to the parent tissue [29]. Organoid production involves culturing primary cells of normal (stem cells) or cancerous tissue and subjecting them to an environment that is near-native to in vivo conditions, thus allowing them to organize themselves on their own. This self-organization seems to occur in a cellular system that lacks an orderly structure and is guided spatially to rearrange by autonomous mechanisms in culture media supplemented with specific molecules representative of the native tissue supplemented with a matrix [30]. This enables the organoids to copy the genotype and phenotype of the parent tissues much more efficiently than any other type of 3-D cell culture. It has also been found that organoids are able to stimulate the formation and secretion of extracellular vesicles similar to in vivo tissues [31]. The matrix that is predominantly used is the basement membrane extract (BME) or one of its commercially available variants, such as Matrigel, while the growth factors vary according to the tissue-specific type of organoid cultured. Organoids have been used for various purposes, including the study of TME, for personalized medicine and immune-therapeutics, and modelling of carcinogenesis to name a few. Thus, it seems to be the most promising avenue to pursue research in translational oncology.

There are many aspects of OSCC that are still not fully understood, particularly in the area of oral carcinogenesis. In vitro 3-D cell cultures have now been used in the field for a number of years and the goal of this review was to critically assess the studies that had utilized 3-D cultures and to identify knowledge gaps in the literature. For this purpose, we selected a scoping review approach [32] to understand how 3-D cell cultures have been used to study the biology of normal and dysplastic oral mucosa and oral carcinoma tissues. In addition, the review also investigated the various culture conditions used for spheroid, organoid, and organotypic culture in relation to cells of the oral cavity.

## 2. Materials and Methods

### 2.1. Study Design

This review was performed according to the Preferred Reporting Items for Systematic Reviews and Meta-Analysis Extension for Scoping Reviews (PRISMA-ScR) [33]. The final protocol was registered prospectively with the Open Science Framework on 17 December 2019 (https://osf.io/x2vef). Electronic databases, such as PubMed, Web of Science, and Scopus, were searched until August 2020 using a combination of “MESH terms” (Appendix A). In addition, references were hand-checked from bibliographies in relevant articles and included in this review.

### 2.2. Inclusion and Exclusion Criteria

The selection process based on the PICOS model was as follows. The population (P) was the keratinocytes (cell lines or primary cells) derived exclusively from the oral cavity (normal, dysplastic, or carcinoma) mucosa. The intervention (I) was the use of 3-D cell cultures, such as spheroid or organotypic or organoid cultures. There were no comparators (C) in our study and the primary outcome (O) measured was the purpose for which 3-D cultures were used in our population. The secondary outcomes measured were the culture conditions for all three methods of 3-D cell culture. The study design (S) included all original studies published in the English language where the full text was available. The site of derivation of cell lines was checked and confirmed in Cellosaurus (https://web.expasy.org/cellosaurus/). Review articles, abstracts from conference proceedings, and studies not associated with cells from the oral cavity, such as tonsil, pharynx, larynx, or other head and neck sites, were excluded. Any studies associated with animal cells or xenografts were also excluded as they do not fall into the scope of this review. Studies associated with viral oncogenesis were also excluded as they are representative of oropharyngeal SCC rather than oral SCC [34].

### 2.3. Study Selection

To ensure high inter-rater reliability, a training exercise was conducted before starting the screening process. A custom questionnaire was designed according to the inclusion and exclusion criteria. This was developed and tested on a random sample of 50 manuscript titles and abstracts (i.e., level 1 screening) by all team members. The same exercise was repeated for the screening of full-text articles (i.e., level 2 screening). Subsequently, two members (RS and OK) screened all full-text articles for inclusion, independently, for level 1 and 2 screening. Inter-rater discrepancies were resolved by discussion or by a third adjudicator. 

### 2.4. Risk of Bias Assessment or Quality Appraisal

Since this is a scoping review conducted to identify gaps in knowledge, no risk of bias assessment or quality appraisal was carried out, in accordance with the manual published by the Joanna Briggs Institute [35].

### 2.5. Synthesis of Results

All data was broadly divided into three categories: spheroids, organotypic raft cultures, and organoids. In line with the aim of the study, for all categories, primary and secondary outcomes were tabulated. In addition, the year of publication of all studies was included to map the first utilization of 3-D cultures in relation to other tissues of the body.

## 3. Results

### 3.1. Data Items and Extraction

Our search strategy returned a total of 199 studies of which the full text of 178 articles was screened for eligibility criteria. A total of 59 articles were excluded with reasons explained in Appendix A. Finally, 119 articles were used for data extraction and measured for primary and secondary outcomes. Figure 3 summarizes the screening process showing the number of articles passing through each step.

### 3.2. Characteristics of the Included Studies

The retrieved articles were stratified into three categories with 53 articles related to spheroids, 62 associated with organotypic raft cultures, and 4 articles related to organoids.

### 3.3. Spheroids in OSCC

The first spheroid cultures from OSCC cell lines were developed in 1989. The most common method of generating spheroids in OSCC (Appendix A) was the liquid overlay method, with 33 of the studies (62%) utilizing ultra-low attachment dishes or wells to culture spheroids, which were either commercially available plates or coated with poly-HEMA or agarose. Three studies (5.5%) used the hanging drop method. Six studies (11%) used the matrix embedded method to produce spheroids and four studies (7.5%) used the cells above the matrix method. One study (2%) used both the hanging drop and the commercially available ultra-low attachment plates and two studies (4%) did not mention the technique they used to generate spheroids. Four studies (7.5%) used novel methods of spheroid generation, such as spheroid catch, CELLine AD1000 flask, and aerosol-based microencapsulation systems, and a 3-D-printed microfluidic perfusion device. None of the studies used the agitation-based techniques, such as the rotating wall vessel or magnetic levitation technique, to culture cells as spheroids. Fifty-two of the studies used OSCC cells and one study used normal oral keratinocyte cell lines (OKF-6 and HOK). None of the studies used dysplastic cells to be cultured as spheroids. Of the 53 studies, only 3 used a co-culture system along with cancer cells. The purpose of utilization of the spheroids for research in OSCC is shown in Figure 4.

### 3.4. Organotypic Cultures of Cells Associated with the Oral Cavity

The first organotypic raft cultures from oral keratinocytes were produced in 1996. The culture conditions used in the majority of studies (42 of 62) included a co-culture system (Appendix A). Of these studies, 38 used fibroblasts from various sources whereas one study used both fibroblasts and human lymphatic endothelial cells, while the remaining three studies each used peripheral blood monocytes, leukemia cells, or bone marrow-derived mesenchymal cells. The most common scaffold used was collagen I (30 studies) from different sources, followed by human uterine leiomyoma discs (15 studies). Variants of de-epidermized dermis/skin were used in seven of the cultures whereas a mixture of collagen and Matrigel was used in three of them. One of the models used a novel method of fibronectin and laminin-coated fibroblasts to produce a 3-D scaffold for cells to grow on and one study used Matrigel alone. One study used both the collagen and human uterine leiomyoma discs, one study used three matrices (collagen, Matrigel, and human uterine leiomyoma disc), one study used both collagen and a 3-D bio-printed bone scaffold, and another study used decellularized de-epidermized dermis and collagen. The purposes for which organotypic models have been used are summarized in Figure 5.

### 3.5. Organoids in OSCC

Organoids from OSCC cells were first generated in 2018. There has been a total of four studies of which three have used primary cells and one used an OSCC cell line. Only one study used primary keratinocytes from normal mucosa and there were no studies that utilized dysplastic cells to generate or propagate organoids. Three of the studies used variants of BME, such as Matrigel or growth factor reduced Matrigel, as a scaffold, and one study used collagen. The growth factor supplementations were quite different in all the studies and are shown in Table 2. Organoids were used as models to study oral tissues, drug testing, TME, or hallmarks of cancer.

## 4. Discussion

Cell culture models are an important pre-clinical tool for oncology researchers. Animal studies are also an important component of pre-clinical investigation but have a number of disadvantages, including ethical concerns, expense, handling, and the difficulty of translating research findings from non-human models to the clinic [40]. As cell culture has advanced from the traditional monolayer cultures to 3-D approaches, it has been able to mimic the human TME much more effectively and represent a promising tool to investigate various aspects of cancer and pre-cancer in the laboratory.

There are still substantial gaps in understanding the process of oral carcinogenesis. It is known that tobacco and alcohol are important risk factors in the development of OSCC, and that many OSCCs are preceded by oral potentially malignant disorders (OPMDs). In a clinical scenario, patients with OPMDs who present to the clinic typically undergo a biopsy and a pathologist assesses the presence and grades epithelial dysplasia as mild/moderate/severe or low/high risk based on the cellular and architectural changes in the epithelium [41,42]. Severe or high-risk dysplastic lesions have been shown to carry the highest risk for malignant transformation, but more than 50% of these lesions do not become malignant and the reasons for this remain unclear [43]. Dost et al. noted that grading OPMDs based on histology might not be sufficient to predict which of the lesions undergo malignant transformation clinically [44]. These factors highlight the importance of understanding the molecular development of OSCC. This understanding would allow the identification of potential biomarkers that could be used to predict the malignant transformation of lesions or selectively drug dysplastic cells.

Recent advances in 3-D cell culture approaches, such as spheroids, organotypic cultures, and organoids, provide an excellent platform to investigate unanswered questions related to oral carcinogenesis. In this review, we sought to analyze how researchers have used these platforms in OSCC research, specifically in relation to oral carcinogenesis. One interesting finding is that oral cancer research has been slow to adopt 3-D cellular techniques compared with cancer research generally. Multicellular tumor spheroids came into use as early as 1970 [45], yet the first spheroids from OSCC cells were cultured almost 20 years later in 1989 by Schwachofer et al. [46]. Organotypic raft cultures displayed a similar finding, with the first such cultures developed in 1984 [47] from epidermal cells of the skin but not applied to oral keratinocytes until 1996 by Eicher et al. [48]. Comparable observations were made in organoids with the first organoids from intestinal cells cultured in 2009 by Sato et al. [49] and the first organoids from OSCC cells cultured in 2018 [36,37]. All 3-D culture technologies were implemented in OSCC some 10–20 years after their initial introduction, which suggests that an emphasis on rapid implementation of novel 3-D techniques has great potential to enhance oral cancer research.

The most common method that has been utilized to generate spheroids is the use of ultra-low attachment plates or wells (liquid overlay method) to generate 3-D spheroids. This method provides a relatively simple and inexpensive way of generating spheroids by encouraging cell–cell interaction, which likely explains its popularity [50]. Hagemann et al., in their study using OSCC cells, compared the hanging drop and liquid overlay method among the suspension-based models and found the latter to be superior as the average growth rate of spheroids was almost double when compared to the hanging drop method [51]. Interestingly, only Dennis et al. used normal oral keratinocyte cell lines [52] and none of the studies used dysplastic cells to be cultured as spheroids. All other studies used OSCC cells, and a majority (23) of them were used as a tumorosphere forming assay or to assess cancer stem cell properties of OSCC cells, which is evaluated by the ability of cells to form a 3-D spheroid structure [53]. Our analysis indicates that the spheroid models in OSCC have been used to investigate various aspects of TME and the hallmarks of cancer, or as drug-testing platforms for OSCC cells but have not been used to study carcinogenesis or the microenvironment of dysplastic tissues. 

Organotypic raft cultures seem to be the most popular of the 3-D cell culture methods as more than half of the included studies (62/119) used this model. This could be due to the fact that this model suitably recapitulates keratinocyte differentiation, which is a critical feature in epithelial cancer, such as OSCC. The culture techniques used in most of the studies include the use of fibroblasts from various sources in a scaffold containing collagen or human uterine myoma discs. This reinforces the fact that fibroblasts and scaffolds play a major part in forming the TME of OSCC and have been successfully modelled in a 3-D in vitro environment [54,55].

Similarly to spheroids, a major proportion of the studies used these models in some way to mimic the TME or to understand the invasive capabilities of OSCC cells. This appears to be related to the construct of the model, which accurately represents the in vivo 3-D TME by including both fibroblasts and a matrix component. These culture models have also been used to test various pharmaco-, chemo-, or radiotherapeutic modalities in OSCC, and, most interestingly, the features of dysplastic cells. This is in contrast to studies using spheroid cultures, with organotypic cultures representing the only 3-D cell culture method in which oral dysplastic cell lines have been cultured. AbdulMajeed et al., Chaw et al., Dalley et al., and Vigneswaran et al. cultured normal, dysplastic, and OSCC cell lines as models of their respective tissues and found that they were useful in identifying biomarkers to predict malignant transformation of dysplastic lesions [56,57,58,59]. Yoo et al. attempted to create an in vitro model mimicking the natural progression of OSCC from normal keratinocytes by transforming them using a chemical carcinogen in traditional monolayer cultures and then culturing them as organotypic raft cultures [60]. Thus, in spite of organotypic cultures being used primarily for studying invasiveness, they have also been used to model dysplastic tissues, and to study the natural progression of OSCC from normal to malignancy, which was absent in the spheroid models. 

Organoids are the latest technology in 3-D cell culture and have been very recently applied to model tissues associated with the oral cavity. Driehuis et al., Tanaka et al., and Zhao et al. used primary cells to model tissues of normal and OSCC mucosa and were successfully able to recapitulate the genetic and phenotypic make-up of the parent tissues [37,38,39]. All the studies associated with organoids were used as models representing their parent tissues or utilized to study OSCC TME. Driehuis et al. [38] showed that organoids could potentially be used for personalized medicine. They tested a range of inhibitors as targeted therapies on tumor organoids and found that selective sensitivity could be observed when using mutation-specific inhibitors. Similarly, Lee et al. showed a promising response using bladder cancer patient-derived organoids, when inhibitors specific to the patient’s mutational profile were combined with traditional chemotherapeutic agents [61]. Kim et al. found that *BRCA2* mutant-lung cancer organoids showed a good response to olaparib whereas *EGFR* mutant and *EGFR*-mutant*/MET*-amplified organoids showed better response to erlotinib and crizotinib, respectively [62]. All these efforts can be considered as proof-of-concept to incorporate personalized medicine in OSCC treatment via patient genetic testing and subsequent screening of drugs using organoid technology in the lab to identify the most appropriate treatments. This is especially important for patients with carcinogen-associated cancers, such as OSCC, whose mutational burden is very high and profiles are quite variable. Organoids appear to be useful in immuno-oncology as observed by Cattaneo et al., who successfully established protocols to co-culture autologous T lymphocytes and tumor organoids in lung and colon cancer, paving the way for research in precision immunotherapy [63]. 

Organoid technology, as with spheroids, represent a promising approach to modelling oral carcinogenesis, and have been used for this purpose in other cancers. Drost et al. introduced sequential mutations in normal intestinal stem cell-derived organoids using CRISPR/Cas9-mediated gene editing to establish tumor organoids, which formed invasive tumors upon re-implantation in mice [64]. Similarly, Naruse et al. used normal mouse liver and lung organoids and treated them in vitro with chemical carcinogens and found them to be tumorigenic in nude mice [65]. These methods of introducing mutations by chemical carcinogens or by introduction of oncogenic loci have been termed the ‘bottom-up’ approach of studying oncogenesis using organoids. This method combines the advantages of the study of carcinogenesis by traditional 2-D cultures and in vivo animal models by allowing facile genetic manipulation and by keeping the complexity of the 3-D human TME [66]. These studies show that 3-D in vitro organoids could be useful in understanding early molecular changes that occur in oral carcinogenesis and potentially further elucidate the role of different agents or inhibitors in chemoprevention. 

Organoids have also been used to study various aspects of TME, such as invasion, and to identify specific subpopulations of cells involved in invasive tumors. Cheung et al. showed that malignant breast cells expressing keratin-14 and p63 led to the collective invasion of the tumor cells [67]. Such studies can be applied to oral cancer with cells obtained from early invasive cancer patients or from transformed dysplastic tissues cultured as organoids to observe their invasive behavior. This could potentially be useful in identifying subpopulations of cells involved in oral carcinogenesis. Investigation of niche factors involved in carcinogenesis has been performed using organoids. For example, Fujii and colleagues showed that colorectal tumor organoids cultured with selective inhibition of different niche factors demonstrated involvement of specific molecules, such as EGF, in carcinogenesis and the niche factor independence was observed in the transition from adenoma to carcinoma. They combined these results with sequencing data to illustrate that the synergistic effect of mutations in RAS/MAPK and PI3K pathways led to EGF independent growth of tumor organoids [68]. Studying oral oncogenesis in vitro may require a complicated setup involving the presence of normal, dysplastic, and cancerous tissue in the same model. Recent technologies, such as 3-D bio-printing, have allowed chimeric organoids to be cultured consisting of both normal and cancerous breast tissue co-printed in the same plane [69]. Such studies provide evidence that organoid technology combined with next-generation sequencing and bioprinting could potentially be used to study oral carcinogenesis with a relatively better microenvironmental setup when compared to other forms of in vitro cell culture models. A major advantage of organoids is that they allow the creation of biobanks and can be propagated from cryopreserved primary cells or from genetically/chemically transformed organoids. This is advantageous for clinician researchers investigating oral carcinogenesis. It is challenging to identify if and when a dysplastic lesion will undergo malignant transformation. Thus, cryopreserved dysplastic cells that later clinically turned malignant when propagated as organoids might potentially be useful in understanding the key molecular mechanisms of neoplastic transformation. 

Whilst organoids present a promising platform for modelling and studying oncogenesis, they are not without disadvantages. A major issue is that the most commonly used culture matrix, Matrigel, is derived from mouse sarcoma and cannot fully recapitulate the human tissue microenvironment. This is being addressed in a number of studies, for example, by Mollica et al., where matrices were prepared from decellularised human breast tissue and 3-D-printed organoids and tumoroids generated [70]. A further disadvantage of organoid culture is that not all primary cells can readily be cultured as organoids. Driehuis et al., in their study, found that they could successfully propagate 60% of the primary OSCC tissue samples as organoids, although this may be improved by further optimization [38]. Organoid generation is also a fairly costly process and it would be important to assess the benefits in patient care in a scenario where organoid growth and drug screening is used to guide cancer treatment. Biobanks of organoids provide an important resource for research and development and seem to maintain the characteristics of the primary tumor at the DNA level [71]; however, it is unclear whether intra-tumor heterogeneity is captured. Similarly, whilst attempts at passaging of organoids appear successful, it will be important to fully investigate how long these cultures can be maintained before ‘clonal drift’ occurs [72]. Given the wealth of organoid studies in the pipeline, these challenges will undoubtably be addressed in the near future and create a positive impact on translational oncological research. Based on the above data, it is clear that 3-D cell culture models have not yet been utilized to their full potential in OSCC research, especially in understanding the natural progression of the disease. This is an important aspect for future research in oral oncology. In vitro modelling of oral carcinogenesis and dysplastic tissues and testing of chemo-preventative agents are currently lacking, and it would appear that organoids may be a platform that can fill this gap as a viable replacement for animal models and to benefit translational research. 

Future research in the use of 3-D cell cultures for OSCC should also attempt to incorporate recently and rapidly advancing technologies in engineering that allow the use of microfluidic organ-on-a-chip platforms. These models mimic blood vessels, which facilitate gas and nutrient exchange in a much more physiological and efficient manner. They also allow for controlled fluid movement containing biomolecules, cells, microorganisms, or other chemical compounds and can be used to produce a customized microenvironment for dysplastic and malignant oral cells [73,74]. Joint use of these technologies along with 3-D bio-printing can tremendously improve the complexity of in vitro models, as there is potential to combine a physiologically relevant environment with personalized patient cell-laden materials to model a precise patient-specific microenvironment in the laboratory [75]. Another interesting advancement in engineering is the introduction of ‘4-D’ bio-printing. The integration of an additional dimension ‘time’ to existing 3-D printed models allows the structures to change over time when an external stimulus is applied. This can be achieved by multiple mechanisms, but the method most relevant to oral oncogenesis is the use of magnets and incorporation of iron oxide loaded nanoparticles in the target component of the model [76]. This could allow architectural changes associated with dysplasia, such as formation of ‘tear-drop or bulbous’ rete ridges, to be mimicked in the lab. Experimentally, this would involve subjecting the 3-D bio-printed basement membrane to magnetic fields and allowing it to change its shape when external stimulus, such as carcinogens, are applied. This can further broaden avenues to study the detailed process of transformation from normal to dysplasia to malignancy in vitro.

Overall, 3-D cell culture models are moving closer to achieving complete biomimicry although there remain a number of areas to be addressed before this is attained. The environment created in these models still does not entirely represent the TME as there are many elements, including microorganisms, blood vessels, nerves, and various other cells, cytokines, and growth factors, that cannot yet be incorporated within them. 

## 5. Conclusions

Three-dimensional cell culture technologies are gaining significant importance in translational research as they can bridge the gap between in vitro models and the in vivo tissue environment. Spheroids, organotypic, and organoid cultures have improved in complexity and physiological relevance and have thereby become more available for researchers as a platform to model carcinogenesis and for development of personalized medicine protocols. These technologies have been tested in cancer research more broadly for personalized medicine therapies, but there is greater scope for them to be used in the field of oral carcinogenesis and chemoprevention.

## Figures and Tables

**Figure 1 ijms-21-09520-f001:**
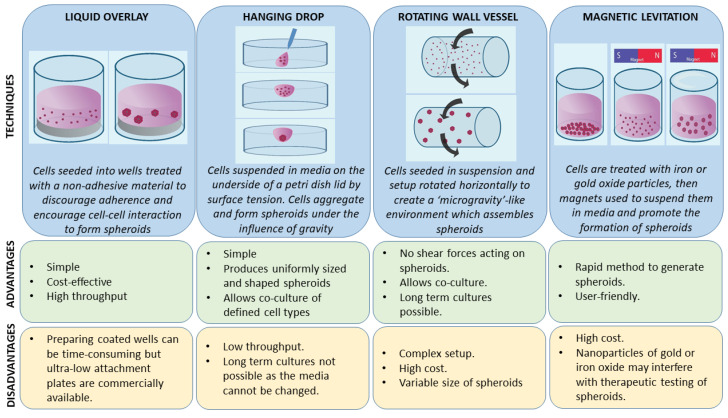
Principles, advantages, and disadvantages of various tumor spheroid model techniques (suspension based).

**Figure 2 ijms-21-09520-f002:**
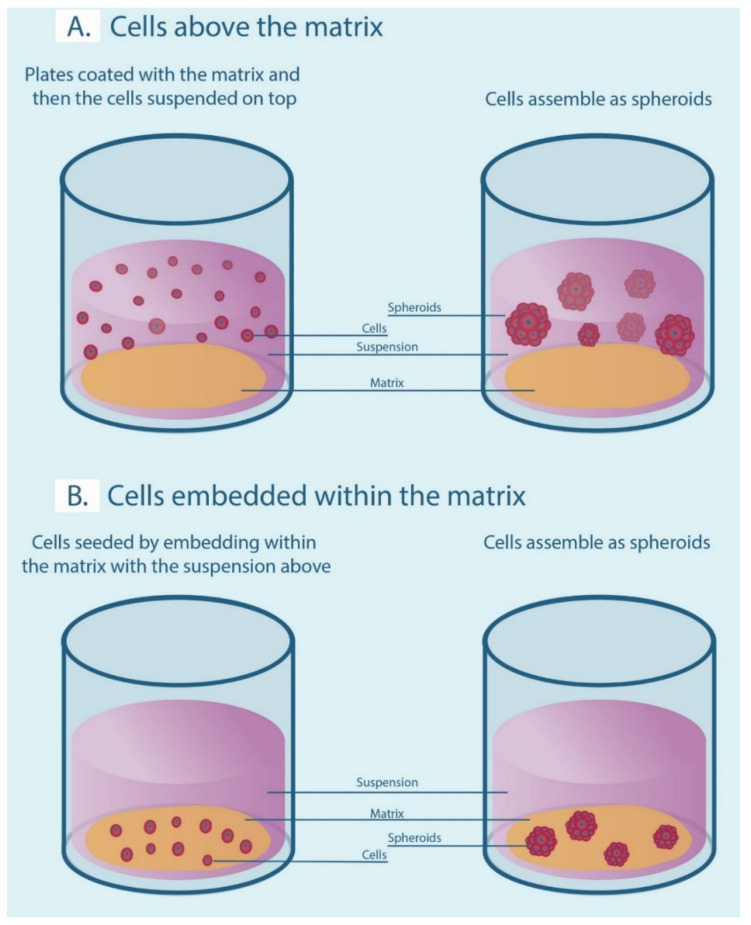
Matrix-associated methods for generating spheroids. (**A**) Cells are seeded above the matrix-coated plates in a suspension. (**B**) Cells are embedded within the matrix to form spheroids.

**Figure 3 ijms-21-09520-f003:**
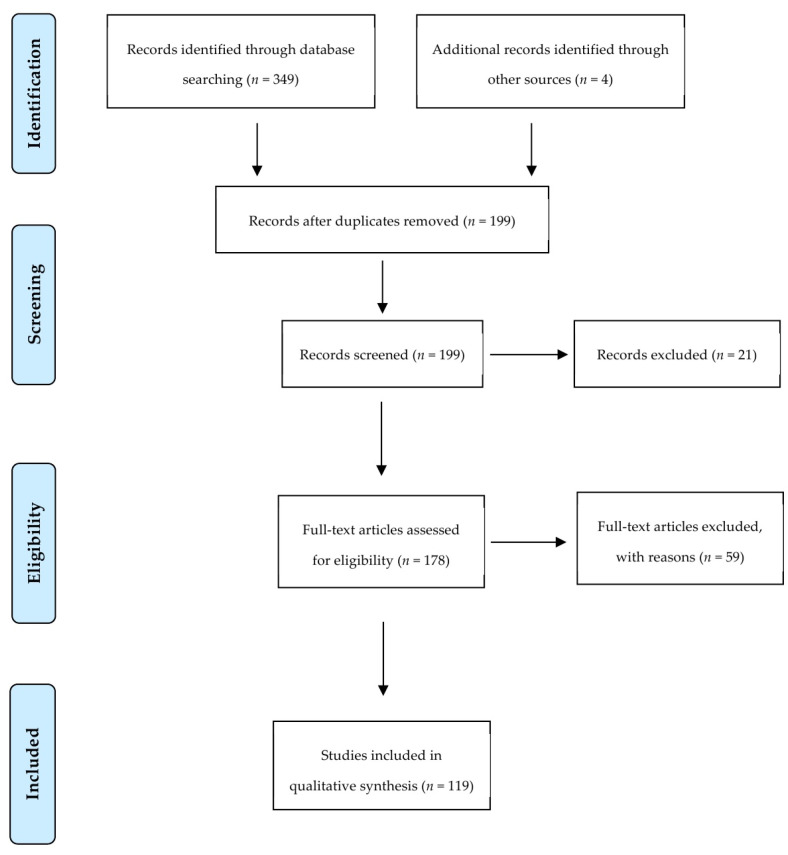
Preferred Reporting Items for Systematic Reviews and Meta-Analysis flow diagram of screened studies.

**Figure 4 ijms-21-09520-f004:**
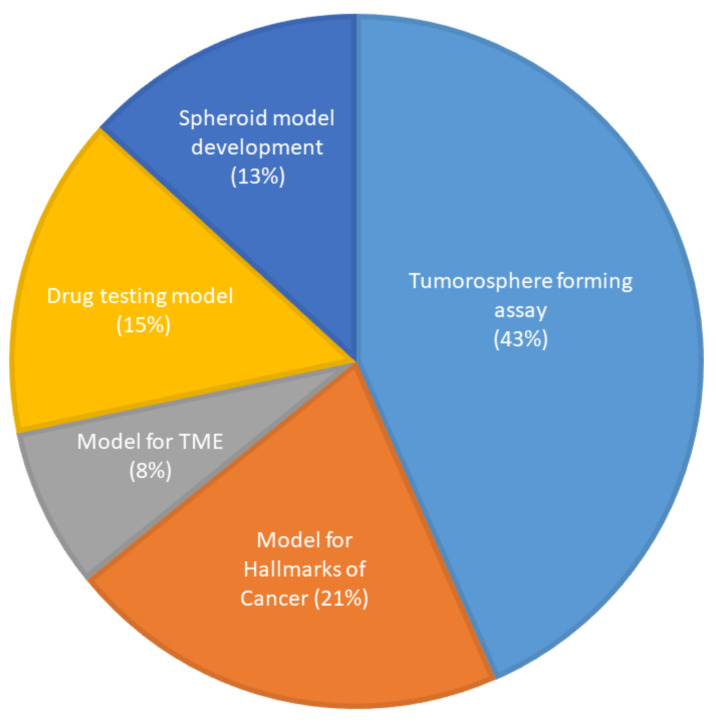
Use of spheroid culture in oral cancer research (*n* = 53).

**Figure 5 ijms-21-09520-f005:**
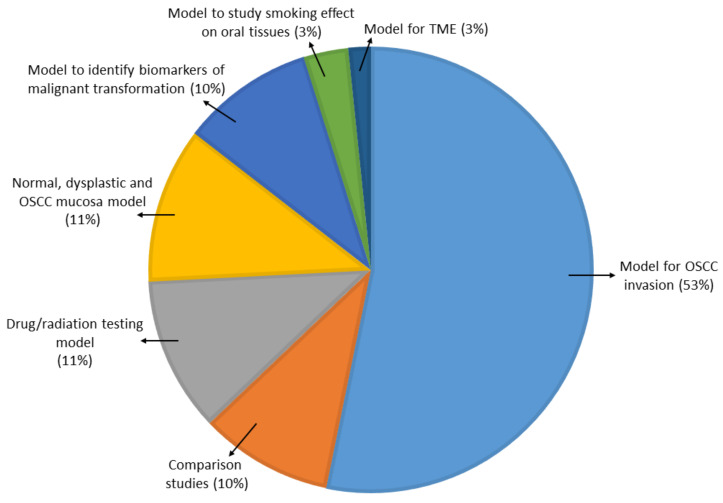
Use of organotypic models in oral cancer research (*n* = 62).

**Table 1 ijms-21-09520-t001:** Natural matrices used in three-dimensional 3-D in vitro cancer models.

Matrix	Features	Advantages	Disadvantages
Matrigel [19]	Derived from Engelbreth–Holm–Swarm (EHS) mouse sarcoma cells Consists of type IV collagen, laminin, tenascin and other proteoglycans and growth factors associated with basement membrane Gold standard for matrices in 3-D cancer models	Ability to promote cell-cell and cell-matrix interactions Can be used with a wide variety of cancer cells and for co-culturing of different types of cells	Batch to batch variability Many undefined factors (overcome by using growth factor reduced Matrigel) Does not contain Type 1 collagen and hyaluronic acid which are important constituents of TME
Fibroblast derived matrices [20]	Derived from cancer-associated fibroblasts	Can mimic tumor microenvironment of advanced carcinomas Near natural extracellular matrix	Batch to batch variability Lengthy preparation time Only a limited thickness of the matrix can be achieved
Type 1 Collagen [21]	Derived from a variety of sources such as bovine skin, rat tail and human sources	Mimics tumor environment very closely Used for studying a variety of hallmarks of cancer Physical and mechanical properties very similar to the extracellular matrix of cancers	Batch to batch variability Difficulty in retrieving spheroids after experimentation for analysis Microstructure of collagen might be different if sourced from non-human sources
Silk [22]	Relatively easily derivable scaffold Various silkworms can be used as sources	Biocompatible Provides site of cancer cell attachment and growth Good mechanical properties	Does not replicate the tumor microenvironment
Alginate [23]	A biodegradable hydrogel derived from plant sources	Good mechanical properties Stiffness of the matrix can be controlled to simulate the extracellular matrix of specific tumors Pore size also can be controlled	Non-adhesive to cells (which can be overcome by adding components that can promote adhesion)
Hyaluronic acid [24]	Essential component of the tumor micro-environment Mainly derived from bacterial sources	Easy manipulation Interacts with cancer cells to form spheroids	Relatively poor mechanical properties Does not simulate all aspects of the tumor microenvironment

**Table 2 ijms-21-09520-t002:** Studies utilizing organoids as a 3-D cell culture model (*n* = 4).

Study No.	Authors (Year)	Type of Cell Used	Site/s Obtained	Culture Conditions	Purpose of the Study
1	Tam et al., (2018) [36]	Cancer cell line (PCI 13)	Oral cavity	Rat tail collagen, DMEM media containing 10% FBS and IFNγ (50 U/mL) ± doxycycline	Hallmark of cancer and TME—cell proliferation
2	Tanaka et al., (2018) [37]	Primary OSCC cells	Buccal mucosa, tongue, floor of mouth, gingiva	Matrigel, StemPro hESC serum-free medium supplemented with bFGF	Model for OSCC tissue Drug testing model
3	Driehuis et al., (2019) [38]	Primary OSCC cells	Oral cavity	BME, Advanced DMEM +/+/+ supplemented with1 × B27, 1.25 mmol/L N-acetyl-l-cysteine, 10 mmol/L Nicotinamide, 50 ng/mL human EGF, 500 nmol/L A83-01, 10 ng/mL human FGF10, 5 ng/mL human FGF2, 1 μmol/L Prostaglandin E2, 3 μmol/L CHIR, 1 μmol/L Forskolin, 4% R-spondin, and 4% Noggin	Model for normal and cancerous oral mucosa Drug testing model
		Primary normal keratinocytes			
4	Zhao et al., (2019) [39]	Primary OSCC cells	Oral cavity	Growth factor reduced Matrigel, DMEM/F12 supplemented with 1× N2, 1× B27, 50 ng/mL human EGF, 10 nmol/L Gastrin, 500 nmol/L A83-01	Hallmark of cancer and TME—effect of lactate on cancer stem cell property

bFGF—Basic fibroblast growth factor, DMEM—Dulbecco’s Modified Eagle Media, EGF—Epidermal growth factor, BME—Basement membrane extract, FGF—Fibroblast Growth factor, IFN—Interferon, OSCC—oral squamous cell carcinoma, TME—Tumor microenvironment.

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
