# Peer review of "Three-Dimensional Cell Culture Models to Investigate Oral Carcinogenesis: A Scoping Review"

_ijms, 2020, doi:10.3390/ijms21249520_

Round 1
Reviewer 1 Report
The format of this review paper is more like an original article, which is inappropriate to this reviewer. The suggested outlines by this reviewer:
Abstract
Introduction
Spheroid culture
Organotypic culture
Organoid culture
Conclusion (including unsolved problems and future applications)
Author Response
Authors’ response: we thank the reviewer for the suggestion. However, our current paper aimed to study the 3D cell culture models in investigating oral carcinogenesis using a scoping review style. Therefore, we followed the guidelines of the PRISMA extension for scoping reviews in our reporting (Tricco, AC, Lillie, E, Zarin, W, O'Brien, KK, Colquhoun, H, Levac, D, Moher, D, Peters, MD, Horsley, T, Weeks, L, Hempel, S et al. PRISMA extension for scoping reviews (PRISMA-ScR): checklist and explanation. Ann Intern Med. 2018,169(7):467-473. doi:10.7326/M18-0850.).
In their proposal, the scoping review would have a structured format similar to original research article and this explains why our current manuscript has a structure similar to original article. Nonetheless, the results section includes paragraphs outlining the subheadings that the reviewer suggested.
Reviewer 2 Report
The authors have put together an excellent review of 3D cell culture models of OSCC, which will be useful to scientists developing methods to treat and cure this cancer.
There are a few minor but necessary corrections:
The citations in the text should come before the period at the end of the sentence.
The journal names need to be abbreviated according to the IJMS instructions. I suppose the abbreviated names do not need a period after names, although this is not delineated in the instructions.
The article titles should be all lower case, except for the first letter and proper names (e.g refs 8, 30, 31, 33, 36, 42, 73 (e.g Joanna Briggs Institute should be capitalized).
In ref 68, "USA" should be added after PNAS.
Author Response
We thank the reviewer for taking the time and efforts to appraise our work. We revised the manuscript per your comments and have used track changes
The authors have put together an excellent review of 3D cell culture models of OSCC, which will be useful to scientists developing methods to treat and cure this cancer.
Authors’ response: thanks for kind comments
There are a few minor but necessary corrections:
The citations in the text should come before the period at the end of the sentence.
Authors’ response: Changes have been made according to the suggestion of the reviewer.
The journal names need to be abbreviated according to the IJMS instructions. I suppose the abbreviated names do not need a period after names, although this is not delineated in the instructions.
Authors’ response: All references have been checked and amended in accordance with IJMS format.
The article titles should be all lower case, except for the first letter and proper names (e.g refs 8, 30, 31, 33, 36, 42, 73 (e.g Joanna Briggs Institute should be capitalized).
Authors’ response: thanks for careful check. All references have been checked and amended in accordance with the IJMS format.
In ref 68, "USA" should be added after PNAS.
Authors’ response: amended.
Reviewer 3 Report
This is a sound, well written systematic review paper. To me it looks like all relevant papers are mentioned and fairly analysed. I consider thus that the review contains important information for those interested in in vitro modeling of OSCC and deserves to be published.
Author Response
we thanks the reviewer for taking the time to read and appraise our work. we are also thankful for commending our work and kind comments